# Elovanoids, a Novel Class of Lipid Mediators, Are Neuroprotective in a Traumatic Brain Injury Model in Rats

**DOI:** 10.3390/biomedicines12112555

**Published:** 2024-11-08

**Authors:** Nicolas G. Bazan, Andre Obenaus, Larissa Khoutorova, Pranab K. Mukherjee, Bokkyoo Jun, Rostyslav Semikov, Ludmila Belayev

**Affiliations:** 1Neuroscience Center of Excellence, School of Medicine, LSU Health New Orleans, New Orleans, LA 70112, USA; lkhout@lsuhsc.edu (L.K.); pmukhe@lsuhsc.edu (P.K.M.); r.semikov@audubonbio.com (R.S.); 2Division of Biomedical Sciences, University of California Riverside, Riverside, CA 92507, USA; andre.obenaus@medsch.ucr.edu; 3Audubon Bioscience, Houston, TX 77021, USA

**Keywords:** omega-3 fatty acids, T2 mapping, diffusion tensor imaging, white matter, fluid-percussion injury

## Abstract

Background: In the United States, traumatic brain injury (TBI) contributes significantly to mortality and morbidity. Elovanoids (ELVs), a novel class of homeostatic lipid mediators we recently discovered and characterized, have demonstrated neuroprotection in experimental stroke models but have never been tested after TBI. Methods: A moderate fluid-percussion injury (FPI) model was used on male rats that were treated with ELVs by intravenous (IV) or intranasal (IN) delivery. In addition, using liquid chromatography-mass spectrometry (LC-MS/MS), we examined whether ELVs could be detected in brain tissue after IN delivery. Results: ELVs administered intravenously 1 h after FPI improved behavior on days 2, 3, 7, and 14 by 20, 23, 31, and 34%, respectively, and preserved hippocampal CA3 and dentate gyrus (DG) volume loss compared to the vehicle. Whole-brain tractography revealed that ELV-IV treatment increased corpus callosum white matter fibers at the injury site. In comparison to treatment with saline on days 2, 3, 7, and 14, ELVs administered intranasally at 1 h and 24 h after FPI showed improved neurological scores by 37, 45, 41, and 41%. T2-weighted imaging (T2WI) abnormalities, such as enlarged ventricles and cortical thinning, were reduced in rats treated by ELV-IN delivery compared to the vehicle. On day 3, ELVs were detected in the striatum and ipsilateral cortex of ELV-IN-treated rats. Conclusion: We have demonstrated that both ELV-IN and ELV-IV administration offer high-grade neuroprotection that can be selectively supplied to the brain. This discovery may lead to innovative therapeutic targets for secondary injury cascade prevention following TBI.

## 1. Introduction

Traumatic brain injury (TBI) is a global public health epidemic, with over 3 million people diagnosed annually in the United States alone [1]. Out of the three types of TBI—penetrating, closed head, and explosive blast [2]—closed head has the highest incidence rate among civilians and can be attributed to sports, falls, or accidents that result in blunt impact. The brain’s normal functioning is disrupted by compression contact and robust and blunt force directly beneath the site of impact, causing abrupt damage to the brain vasculature and neuronal cells. Approximately 90% of all TBIs are mild to moderate, while 80% of TBI cases in the United States are classified as mild (mTBI) [2,3].

Although mTBI often does not cause brain damage directly, it is still accompanied by long-term clinical consequences, including sensory and motor deficits and severe emotional, cognitive, and psychosocial impairment, which cripples essential areas of higher functioning [4]. TBI treatment combines cognitive and occupational therapy and medication to control symptoms such as headaches or anxiety. Currently, no TBI-specific therapies are available [5]. The development of neuroprotective drugs has proven exceptionally difficult due to the blood–brain barrier (BBB) limiting therapeutic penetration. One novel approach to address this challenge is administering drugs intranasally to bypass the BBB [5,6] noninvasively. Using the IN route, drugs can thus be transported along the trigeminal and olfactory nerves directly to the brain from the nasal cavity [6].

Recent evidence strongly indicates that administering omega-3 polyunsaturated fatty acids (PUFAs) and dietary exposure before or after TBI may provide a unique opportunity to enhance the neuronal repair process in experimental TBI [7,8,9]. Omega-3 PUFAs have been studied for decades, showing decreased neuroinflammation and oxidative stress, neurotrophic support, and the activation of cell survival pathways, as well as improved neurological outcomes [10]. After absorption, the omega-3 PUFAs can directly diffuse into cells and interact with fatty-acid-transport proteins [11]. The TBI-caused degradation of membrane phospholipids results in disturbances in cellular membrane functions and contributes to secondary neuronal injury [12]. To date, two omega-3 PUFAs—docosahexaenoic acid (DHA) and eicosapentaenoic acid (EPA)—have the most promising laboratory evidence for their neuro-restorative capacities in TBI [8]. DHA supplementation increases DHA content in the brain after TBI, which may help preserve membrane integrity and enhance neurological function. The role of dietary DHA or EPA supplementation in human neurological injuries remains uncertain despite the laboratory evidence supporting its neuroprotective effects in animal models [13].

Elovanoids (ELVs) are a novel class of homeostatic lipid mediators and derivatives of very-long-chain polyunsaturated fatty acids (VLC-PUFAs, n-3) that we not only discovered but also characterized in recent years [14]. ELVs display neuroprotective bioactivities in both in vitro and in vivo experimental ischemic stroke [15]. The ELVs discovered in our laboratory were stereospecific dihydroxylated derivatives of 32:6n-3 or 34:6n-3 that yielded ELV-N32 and ELV-N34, respectively (Figure 1) [14]. We demonstrated that they had improved behavior and decreased lesion sizes 7 days after the experimental stroke model in rats and protected retinal pigment epithelial cells and photoreceptors [15,16]. Based on our previously shown neuroprotective effect of ELVs in an experimental stroke model, we decided to evaluate ELV therapy at the same doses and time points in an experimental TBI model.

The present study aimed to determine whether treatment with intranasal (IN) and intravenous (IV) ELV administration would be beneficial in a rat model of fluid-percussion injury (FPI). In addition, we examined if ELVs can be detected in brain tissue after IN delivery. The ELV treatments were investigated using a well-established rat model of FPI [17] with the aid of multimodal magnetic resonance imaging (MRI), neurobehavioral assays, and lipidomic analysis. No prior studies have used these novel lipid mediators to treat experimental TBI.

## 2. Materials and Methods

### 2.1. Animals

All animal experiments complied with the National Institute of Health Guide for Care and Use of Laboratory Animals and were approved by the Institutional Animal Care and Use Committee of the Louisiana State University Health Sciences Center (LSUHSC), New Orleans (22 July 2024). Thirty male Sprague–Dawley rats (350–450 g) from Charles River Laboratories (Wilmington, MA, USA) were acclimated for three days before TBI experiments (12 h light/dark cycle) with ad libitum access to food and water. Rats were housed in pairs in a regular rat cage on standard cob bedding with environmental enrichment. Before the surgical procedure, they were fasted overnight with free access to water. All experiments were performed between 8:00 a.m. and 4:00 p.m. The animals were randomly assigned to experimental groups, conducted by researchers blinded to the treatment groups.

### 2.2. Fluid-Percussion Brain Injury (FPI)

The FPI procedure was performed as previously described [17]. Briefly, rats were anesthetized with 3% isoflurane, 70% nitrous oxide, and an oxygen balance. Following endotracheal intubation, the rats were positioned in a stereotaxic frame, and a 4.8 mm craniotomy was made overlying the right parietal cortex (5 mm posterior to the bregma and 4 mm lateral to the midline). Then, a 2.0 mm female luer-lock fitting was positioned over the craniotomy. Instant adhesive glue with an Intra-Set accelerator was used to bond it to the surrounding structures. The scalp was closed with sutures, and the animal returned to its cage and was allowed to recover overnight.

The following day, rats were re-anesthetized with isoflurane, orotracheally intubated, and maintained on 70% nitrous oxide, 3% isoflurane, and a balance of oxygen. The right femoral artery and vein were catheterized for continuous blood pressure monitoring, drug infusion, and periodic blood sampling for arterial gases, pH, hematocrit, and plasma glucose (15 min before TBI and 15 and 45 min after TBI). Mean arterial blood pressure (MABP) was measured via an indwelling femoral arterial catheter and was recorded continuously. Rectal (CMA/150 Temperature Controller, CMA/Microdialysis AB, Stockholm, Sweden) and cranial (temporalis muscle; Omega Engineering, Stamford, CT, USA) temperatures were maintained from 36.5 °C to 37.5 °C before, during, and after TBI. A fluid-percussion device (Model FP 302, AmScien Instruments, Richmond, VA, USA) was connected to the injury tube of the rat’s skull. A moderate TBI was delivered to the right dorsolateral parietal cortex by descending a pendulum, conveying a pressure transient of 2–2.5 atm [17]. The scalp was closed with sutures, and the animal returned to its cage. After the impact, the animals were administered buprenorphine, observed every 30 min up to 6 h. The rectal temperature and body weight were monitored periodically for 14 days after trauma.

### 2.3. Drugs Administration and Experimental Groups

ELVs (Cayman Chemical, Ann Arbor, MI, USA) were dissolved in saline and delivered intravenously and intranasally according to the study protocol. The dose and time of ELV administration were chosen based on previous studies our group conducted [15,16,18].

Series 1—IV: Rats (340–388 g) underwent FPI. ELV 34:6 (300 µg/per rat) or saline treatment (n = 5–6 per group) administered into the femoral vein 1 h after FPI at a constant rate over 3 min using an infusion pump (Model 78-0100, KD Scientific Inc., Holliston, MA, USA). The behavior was evaluated daily for 14 days after FPI, and animals were then perfused for MRI.

Series 2—IN: Rats (365–482 g) underwent FPI. Elov-Mix (32:6, 34:6, and an acetyl form of ELV) and saline administered intranasally (10 µg in 10 µL per nostril; total 20 µg per rat) 1 h and 24 h after FPI (n = 6–7 per group). The behavior was evaluated daily for 14 days after TBI, and animals were perfused for MRI.

Series 3: Rats (365–482 g) underwent FPI. Two separate groups of rats were subjected to the same FPI model, IN Elov-Mix and saline treatments (n = 3 per group), and behavioral evaluation. Rats were sacrificed on day 3, and liquid chromatography-mass spectrometry (LC-MS/MS) was used to detect the ELVs in the anterior and posterior ipsi- and contralateral cortices, subcortex, brain stem, cerebellum, and olfactory tracts.

### 2.4. Behavioral Tests

To confirm the presence of neurological deficits, rats were tested by a blinded investigator on a standardized neurobehavioral battery on days 1, 2, 3, 7, and 14 following TBI [17]. The battery consisted of two tests to evaluate various aspects of neurological function: (1) the postural reflex test and (2) the forelimb placing test.

For the postural reflex test, rats were suspended by the tail for 10 s 1 m above the floor. A score of 0 was assigned to intact rats that extended both forelimbs toward the floor. If the rat flexed one or both forelimbs, a score of 1 was given. A score of 2 was assigned to rats that could not resist the force equally in both directions.

To examine sensorimotor integration, forelimb placing reactions to visual (forward and sideways), tactile (dorsal and lateral), and proprioceptive stimuli were measured for each forelimb. For visual placing, intact rats reached for the table with both forelimbs extended. The lateral movement of the animal toward the table edge allowed sideways visual placing to be assessed. The tactile placing was judged by contacting the dorsal followed by the lateral surface of the rat’s forepaw to the edge of the table. Intact rats immediately placed their paws on the tabletop, while impaired rats were either slow to place or did not. The proprioceptive placing was also assessed by pressing the rat’s paw against the table edge to stimulate limb muscles. A score of 0 was given for normal, immediate placing, a score of 1 if the placing was delayed or incomplete, and a score of 2 indicated absent placing. Total neurological function was scored on a scale of 0–12 (normal = 0, maximal deficit = 12).

### 2.5. Perfusion

After the completion of the neurological testing on day 14, the rats were deeply anesthetized under 5% isoflurane anesthesia in a nitrous oxide and oxygen mixture. A median sternotomy exposed the heart, and the ascending aorta was catheterized via the left ventricle. Perfusion fixation was initiated with a transcardiac infusion of saline for 15 min under a constant pressure of 110 mm Hg. The perfusate was then switched to a 4% paraformaldehyde and continued for 20 min. The rats were then decapitated with a guillotine, and the brain was removed and stored in PBS at 4 °C for MRI imaging.

### 2.6. Magnetic Resonance Imaging (MRI) Acquisition and Analysis

MRI was conducted on day 14 after TBI. High-resolution MRI was undertaken using a 9.4T Bruker Avance (Bruker Biospin, Billerica, MA, USA) running Paravision software (version 5.1, Bruker Biospin) for brain injury and tissue microstructure assessment. The following sequences were performed: (1) T2-weighted image (T2WI) sequence (6500 ms/10 ms of TR/TE), 2 cm field of view (FOV), 50 slices, 0.5 mm slice thickness, a 128 × 128 matrix; (2) diffusion tensor imaging (DTI) sequence in 30 isolinear directions with two b-values (5 b_0_ = 0 and b = 2013.46 s/mm^2^), 12500 ms/36 ms TR/TE, 2 cm FOV, 50 slices, 0.5 mm slice thickness with 4 averages and a 128 × 128 matrix. All matrices were zero-filled to 256 × 256.

The MRI Processor plugin from FIJI (version 1.53f51, NIH, Bethesda, MD, USA) was used to process parametric maps [19]. The DTI quality control, image processing, and region of interest (ROI) delineations were executed using DSI studio (http://dsi-studio.labsolver.org (accessed on 1 June 2020)), Department of Neurological Surgery, University of Pittsburgh, USA). DTI images were corrected for eddy current artifacts and bias field inhomogeneities using N4 bias field correction [20] and eddy correction. The parametric DTI maps were generated using FMRIB’s Diffusion Toolbox from FMRIB’s Software Library (FSL, Version 6.0.7). One hundred and twenty-four regional labels (64 bilateral regions) from the Waxholm Rat Brain Atlas [21] were applied to each animal using the Advanced Normalization Tools (ANTs), and four regional DTI metrics were extracted: fractional anisotropy (FA), radial diffusivity (RD), axial diffusivity (AxD), and mean diffusivity (MD). FA reported the asymmetry of water diffusion, AxD reflected diffusion along the primary direction of water movement, RD represented the diffusion perpendicular to the primary diffusion direction (AxD), and MD measured the average water diffusion in all directions.

Moderate FPI typically does not result in overt lesions, but cortical thinning and regional brain volumes are often reduced. In series 2, the (IN) cortical thickness data were assessed with manual measures using Cheshire image processing software (version 4.3, Parexel, Waltham, MA, USA) at the level of the dorsal hippocampus, extracted and summarized in Excel. These steps revealed significant brain regions and segmentations within the MRI and behavioral data, contributing to a comprehensive analysis.

### 2.7. Brain Sampling and Lipidomic Analysis

Rats (series 3) were sacrificed on day 3. The brains were removed and divided into right and left hemispheres, and the cortex, subcortex (dissected at bregma levels +1.2 and −1.8 mm), olfactory tracts, cerebellum, and brainstem were isolated. Tissues were mechanically dissociated using a manual tissue homogenizer, and lipids were extracted following a modified Folch method described previously [22]. A mixture of internal standards (AA-d8 (5 ng/μL), PGD2-d4 (1 ng/μL), EPA-d5 (1 ng/μL), 15-HETE-d8 (1 ng/μL), and LTB4-d4 (1 ng/μL) was added. LC-MS/MS was performed on a Xevo TQ-S equipped with an Acquity I class UPLC with a flow-through needle (Waters, Milford, MA, USA). Samples were reconstituted in a 2:1 mixture of MeOH:H_2_O and injected into a CORTECS C18 2.7 um 4.6 × 100 mm column (Waters, Milford, MA, USA). The analysis of ELV34 by LC-MS/MS was performed as previously described [15,23].

### 2.8. Statistical Analysis

Statistical comparisons were conducted between groups using GraphPad Prism version 9 for Windows (Graph Pad Software Inc., La Jolla, CA, USA). Welch’s *t*-test was used to compare two independent groups. A repeated-measures analysis of variance (ANOVA) followed by Bonferroni’s test was applied for multiple-group comparisons. Multiple Mann–Whitney tests were used to compare group comparisons within a single individual. All MRI data underwent outlier testing across all regional metrics and were identified using a 1.5 interquartile range-based outlier analysis. For voxel-based metrics, animals were classified as outliers if at least greater than 30% of the 124 regions tested were outliers from the entire group. A *p*-value of 0.05 or less was regarded as statistically significant, and the letter “n” represents the total number of samples per group. All data are presented as the mean ± standard error of the mean (SEM).

## 3. Results

### 3.1. Physiological Variables and Mortality

All measured physiological parameters showed no significant differences between groups. No adverse side effects were observed after ELV administration. One rat died in the saline group on day 7 in series 1, while no animals died in any of the treatment groups. On day 14, the body weight increased by 10% after ELV-IN therapy compared to that by saline treatment.

### 3.2. ELVs Remarkably Improved Behavioral Function 14 Days After FPI

We investigated whether treatments with ELVs administered intravenously (ELV-IV) or intranasally (ELV-IN) affected behavior after FPI. Both ELVs improved the total neurological score of the tactile lateral and visual sideways contralateral forelimb placing reactions (Figure 2a–h).

ELV-IV treatment improved total neurological scores by 20, 23, 31, and 34% on days 2, 3, 7, and 14 compared to treatment with saline (Figure 2b). With ELV-IN, the total neurological score improved by 31, 37, 45, 41, and 41% on days 1, 2, 3, 7, and 14 compared to saline treatment (Figure 2f).

Both ELVs improved tactile lateral and visual sideways placing compared to that of the vehicles (Figure 2c,d,g,h).

### 3.3. ELV-IV and ELV-IN Attenuated Brain Damage, Protected the Integrity of the White Matter (WM), and Preserved Corpus Callosum (CC) Integrity 14 Days After FPI

Figure 3a,b present the representative T2WI images from saline and ELV-treated rats. The saline rats exhibited more edema (hyperintensities) and small hemorrhages (hypointensities) within the external capsule’s white matter than the ELV-IV-treated rats. T2WI abnormalities, such as enlarged ventricles and cortical thinning, were minor in rats treated by ELV-IN compared to those in the saline group (Figure 3b).

We investigated whether treatments with ELVs administered intravenously or intranasally affected white matter connectivity after FPI. Diffusion tensor magnetic resonance imaging (dMRI) was conducted to analyze white matter connectivity. The ELV treatments appeared to protect the integrity of the WM (Figure 3c,d). The fractional anisotropy (FA) maps showed improvements in water directionality after ELV-IV treatment and the corpus callosum (CC) conservation after ELV-IN treatment.

Pseudo-colored FA whole-brain maps (brain and cerebellum) were generated and revealed from ELV-IV- and ELV-IN-treated animals. The contralateral CC was seeded in three contiguous slices, and tractography was performed to assess CC connectivity to the ipsilateral hemisphere (Figure 4). As observed, the connectivity between the hemispheres was significantly reduced in the saline-treated rats (Figure 4a,b). However, robust connectivity between the contralateral and the injured ipsilateral CC was observed in the ELV-IV- and ELV-IN-treated rats (Figure 4a,b).

### 3.4. ELV-IV Recovers Brain Injury Volumes and Improves White and Gray Matter Diffusivity

The whole-brain volume was not significantly altered after either ELV-IV or ELV-IN treatments (Figure 5a). ELV-IV exhibited significant increases in the ipsilateral hemisphere and gray matter volumes (Figure 5b). The white matter volumes also increased but did not reach significance. ELV-IN did not significantly alter white and gray matter or hemispheric volumes (Figure 5c).

The regional volumetric sensitivity and white matter volume changes are presented in Figure 6. Broadly, ELV-IV treatments increased white matter, limbic, and cortical region volumes, while ELV-IN treatment appeared less effective in tissue rescue.

### 3.5. Effect of ELV-IN on AxD

DTI metrics (FA, AxD, RD, and MD; diffusivity) were extracted from 124 brain regions, including the hippocampus (Figure 7, Figure 8 and Figure 9). Decreased AxD measures improvements relative to saline-treated rats and can be considered an axonal health/improvement marker. Decreased MD measures global tissue improvements relative to saline-treated rats. ELV-IV significantly reduced AxD and MD in white and gray matter and the CA1 region of the hippocampus (Figure 7, Figure 8 and Figure 9).

### 3.6. ELV-IN Was Detected in the Brain 3 Days After FPI

We investigated whether treatment with Elov-Mix administered intranasally could be detected in the brain. The experimental design is presented in Figure 10a. ELVs improved the total neurological score by 22, 22, and 31% compared to saline treatment on days 1, 2, and 3 (Figure 10b). The relative abundance of ELVs in brain regions is presented in Figure 10c, where ELV-Mix was found in high levels in the cortex and subcortex ipsilateral to the side of FPI (Figure 10c).

## 4. Discussion

This study examined the therapeutic efficacy of a novel class of homeostatic lipid mediators, ELVs, following TBI in rats. We showed for the first time that both deliveries of ELV-IV and ELV-IN promoted functional recovery, preserved hippocampal volume loss, improved regional diffusion metrics in cortical, subcortical, basal ganglia, and limbic regions, and preserved white matter connectivity. In addition, ELV was detected in the ipsilateral cortex and striatum of ELV-IN-treated rats on day 3.

A prevalent complication of concussion is post-concussive syndrome (PCS) [24], the symptoms of which are characteristically similar to those of the initial concussion. However, PCS symptoms typically last longer, impact a person soon after sustaining a minor head injury, and affect approximately half of all concussed people one month following the initial injury [24]. Focal and diffuse brain injuries are the two categories of primary injuries that result from the immediate impact of different mechanical insults on the brain. Additionally, oligodendrocytes, neuronal axons, and the blood vasculature sustain damage from solid tensile forces, leading to ischemic brain damage and brain edema [25]. The primary characteristic of diffuse TBI is substantial axonal damage, primarily in subcortical and deep white matter tissue, including the corpus callosum and brain stem, which involves axonal transport impairment and axonal cytoskeleton degradation, mitochondrial dysfunction, excitotoxicity, oxidative stress, and apoptotic cell death of glia and neurons [25]. Following TBI, these damages to axons can continue for up to months, indicating a link to delayed secondary pathology of brain edema and hemorrhages [26].

Recently, we demonstrated that the ELV precursors C-32:6 and C-34:6 administered intranasally after experimental ischemic stroke improved behavior, decreased T2WI lesion volume, and increased SMI-71 positive blood vessels and NeuN positive neurons, indicating BBB protection and neurogenesis [16]. Gene expression revealed increased anti-inflammatory and pro-homeostatic genes and decreases in the expression of pro-inflammatory genes in the subcortical area.

The assessment of neurological deficits is an essential indicator of TBI progression or recovery. Although mTBI often does not cause direct brain damage, many mTBI patients appear to recover completely but have post-concussive symptoms, deficits in cognitive and executive function, and reduced cerebral blood flow [27,28]. Several studies reported the beneficial effects of PUFAs on functional recovery after TBI [7]. DHA and EPA, the most promising omega-3 PUFAs, play critical roles in neuroprotection and functional recovery after various traumatic injuries to the brain [8,29]. They are highly enriched in neuronal synaptosomal plasma membranes and vesicles [30]. While prevalent, damage to the plasma membrane is an often overlooked aspect of TBI that can impair neuronal signaling and hamper neurological recovery [30,31]. DHA is abundantly found in the brain, where it is stored mainly in membrane phospholipids, and depletion in brain DHA impairs recovery from TBI [13]. Despite the laboratory evidence supporting omega-3 PUFA neuroprotective effects in TBI experimental models, the role of dietary DHA and/or EPA supplementation remains uncertain. There were positive results of a DHA-related benefit on neurotransmission measured by MRI in one study [32]. In contrast, clinical trials using omega-3 supplements for TBI have continued to fail [10].

Previously, we characterized the bioactivities of a novel class of dihydroxylated lipid mediators, ELVs, which are derived from VLC-PUFAs, n-3 with 32 or 34 carbons, and likely even longer fatty acid chains, presumably through previously unknown pathways [15]. ELVs have structures reminiscent of docosanoids but with different physicochemical properties and alternatively regulated biosynthetic pathways. ELVs enhance the expression of pro-survival proteins that support cell survival from uncompensated oxidative stress, N-methyl-D-aspartate (NMDA) receptor-mediated excitotoxicity (as in epilepsy and other neurological conditions), and experimental ischemic stroke, and at the onset of neurodegenerative diseases [14,15]. We have shown that ELVs are neuroprotective in in vitro and in vivo experimental ischemic stroke [15,16] but have never been tested after TBI.

Key issues influencing efficacy are optimal timing and appropriate ways to deliver ELVs to the brain. We showed that ELV treatments delivered intravenously or intranasally improved neurological recovery throughout the 14-day survival after FPI. Rats treated with ELV-IV 1 h after TBI showed neurological improvements within the second day that continued for 14 days. We observed a significantly improved neurological score from 20% to 34% compared to that of saline treatment. In contrast, when ELV treatment was administered intranasally 1 h and 24 h after TBI, an improved total neurological score was observed within the first day by 31%, which continued for 14 days up to 41% compared to saline treatment. In addition, remarkably improved tactile lateral and visual sideways contralateral forelimb placing reactions were demonstrated by ELV-IN treatments 14 days after TBI.

The remarkable effect of ELV-IN on behavior may explain why we could deliver ELV directly to the brain. If this is correct, the next question we would like to ask is where these EVLs are incorporated into the brain, as this is the first look at EVL treatment in TBI. We used a lipidomic analysis to investigate if ELVs can be detected in the brain. ELVs were found at the highest levels in the cortex and subcortex ipsilateral to the side of FPI 3 days after FPI. Improvement in the neurological score and minimal disruption at the FPI site on T2WI 14 days after FPI may suggest that ELVs preserved tissues at the injury site.

Intranasal delivery has now been targeted as an alternative to invasive delivery to bypass the BBB and directly deliver the therapeutic agent, utilizing pathways along olfactory and trigeminal nerves innervating the nasal passages [33,34,35]. Oral delivery prevents the penetration of drugs into the brain because of delivery through the BBB in 98% of small molecules and 100% of large molecules [36]. The 2% of drugs that reach the brain require high systemic doses to achieve penetration [37]. It was previously demonstrated that DHA made into a microemulsion entered the brain through the intranasal route more than when delivered by the intravenous route [38]. Here, we used a large lipophilic molecule of an ELV to successfully deliver DHA intranasally to the brain to treat TBI.

Magnetic resonance imaging has been increasingly used to diagnose moderate to severe TBI [39]. T2WI is a reflection of the exclusive relaxation properties within tissues, where tissues with excess water (i.e., edema) will have longer or increased T2 values [40]. In contrast, T2 values can be reduced under some conditions, including the presence of blood, increased metabolic demands, or reduced blood flow within the areas of the affected tissue [41]. mTBI often develops persistent neurologic sequelae without overt brain damage [42]. We and others have shown that morphologic and diffusion MRI reports demonstrate clear linkages in histopathological measures, allowing MRI to function as a surrogate readout in treatment studies [39,40]. Our T2WI data demonstrated that saline-treated rats had more edema and small hemorrhages within the external capsule’s white matter than ELV-treated rats. In addition, saline-treated rats had overt cortical lesions, cortex thinning, and enlarged ventricles.

In contrast, no overt lesions, reduced cortex thinning, or overt ventricular changes were observed in ELV-IN-treated rats. ELV treatment did not significantly modify MRI-derived total brain volumes, but the IV treatment group had increased volumes. Significant increases in ipsilateral hemisphere and gray matter volume were observed in the IV treatment group, consistent with neuronal preservation. When regional volumes were examined, the IV group (and, to a lesser extent, IN group) showed significant increases in limbic and cortical volumes, with trending white matter preservation. The hippocampus is a structure that has reduced volumes after TBI [43,44]. Some changes occur in the hippocampus, including inflammation, reactive gliosis (universal reaction to brain injury), and neuronal loss. ELV-IV preserved hippocampal volumes in the CA3 and dentate gyrus (DG) compared to saline treatment. The ELV recovery of hippocampal diffusion MRI metrics was particularly prominent in our study. We found that 14 d post-injury, ELVs significantly reduced the axial, mean, and radial diffusivity measures compared to those of the vehicles in IV-treated rats. These reductions were also present in the IN-treated rats, but there was additional variance in these rats.

The differences between regional changes in the EVL-IV and ELV-IN groups may be attributed to (1) differences in the body weight of the rats between the two groups and (2) known differences between drug delivery to brain tissues, with IV delivery considered more direct. Also, variations in IN delivery have been reported [45], and (3) MRI signatures were only extracted at 14 d post-injury, and there may have been differences in efficacy at different time points. Irrespective of these differences, we conclude that ELVs via either route were efficacious in ameliorating behavioral and MRI perturbations.

Interestingly, TBI results in overt losses of DHA but not other fatty acids, and post-TBI DHA supplementation reduces neuroinflammation [10]. In sports, increased plasma neurofilament light (NfL) was also reversed after DHA treatment [46]. NfL in plasma is considered a marker of neuronal injury, and thus, a corollary is that DHA protects neuronal loss and reduces inflammation [47,48]. A similar mechanism is likely responsible for the improvements we observed using diffusion MRI after ELV treatment. A total of 88% of mTBI patients on MRI have evidence of white matter damage based on measurements of MD and FA. At first, traumatic axonal injury was thought to involve immediate axonal tearing by direct forces associated with traumatic injury. Recently, experiments using anterograde tracers have revealed that a TBI axonal injury is a progressive event involving a focal impairment of axoplasmic transport leading to axonal swelling and the greatest disconnection in the hours to days following TBI [49,50]. White matter connectivity can be mapped using DTI. DHA treatment in juvenile rats after cortical contusion reported decreased edema, the preservation of regional volumes, and normalized white matter radial diffusivity [32]. Our data in adult rats also revealed that ELV treatment significantly reversed axial and mean diffusivity measures consistent with tissue health. This effect was particularly pronounced in the IV treatment group.

In a meta-analysis of studies using DTI, Wallace et al. [51] uncovered evidence of white matter damage in 88% of brain regions after mTBI and 92% after moderate or severe TBI, even in patients in whom injury occurred months previously. Here, we mapped the white matter from the CA1 region of the hippocampus to the subiculum. This connection is critical in memory circuits and permits correlations to behavioral tests. We extracted FA values (or any other DTI metric) from the white matter pathway of interest. FA was considered a measure of white matter integrity. Directionally encoded FA maps demonstrated the preservation of the cortex and improved CC integrity by ELV treatment. Moreover, axial and mean diffusivity were reduced in our study’s CC and fornix, reversing the elevations in vehicle-treated rats. This protection of white matter could have led to the improved behavioral outcomes we observed.

Here, we demonstrated the beneficial effect of ELVs in a well-controlled animal model, the lateral FPI model, which is one of the most clinically relevant models for studying changes in the brain after moderate/severe TBI [52,53]. FPI models mimic closed head injuries in humans and have been widely used to study the molecular and cellular aspects of TBI in humans that cannot be explored in a clinical setting [53,54,55]. The model can reproduce focal and diffuse brain damage in rats, including focal contusion, BBB disruption, local and remote axonal injury, and progressive neuronal loss [56].

### Implications for Clinical TBI Patients

Omega-3 fatty acids have been previously shown to be safe and well-tolerated by patients with several diseases [57]. Lipid emulsions are already given as nutritional support to various patients in clinical settings [58]. Ultra-refined fish oil concentrates containing high quantities of EPA and DHA are commercially available and could be translated into clinical use in TBI patients. Our studies observed no adverse side effects after ELV treatment. The administration of ELV after brain trauma warrants further consideration and clinical investigation as a promising and innovative approach in TBI management.

## 5. Conclusions

We demonstrated here that both deliveries of ELV-IV and ELV-IN attenuated brain damage by improving behavior, protecting the integrity of the WM, preserving CC integrity, recovering brain injury volumes, and improving white and gray matter diffusivity. We also showed that ELVs can be selectively delivered intranasally, as detected in the brain by lipidomic analysis.

## Figures and Tables

**Figure 1 biomedicines-12-02555-f001:**
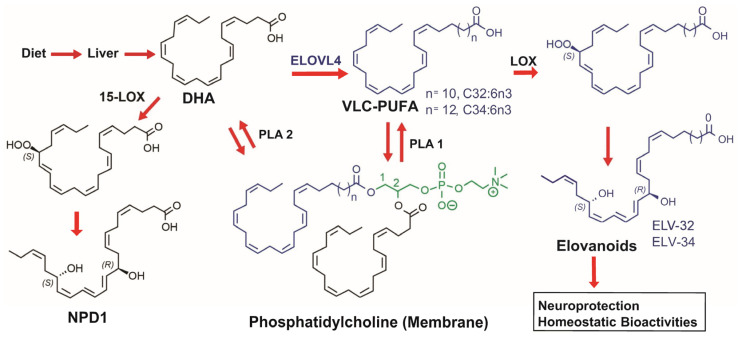
Elovanoid (ELV) biosynthesis. Docosahexaenoic acid (DHA) through ELOVL4 (elongation of very-long-chain fatty acids-4) leads to the synthesis of 32:6n-3, 34:6n-3, and other very-long-chain polyunsaturated fatty acids (VLC-PUFAs). These fatty acids are then esterified at sn-1 of phosphatidylcholine and sn-2 of DHA. Phospholipase A1 (PLA1) releases 32:6n-3 and 34:6n-3, leading to the synthesis of ELV-32 or ELV-34, respectively.

**Figure 2 biomedicines-12-02555-f002:**
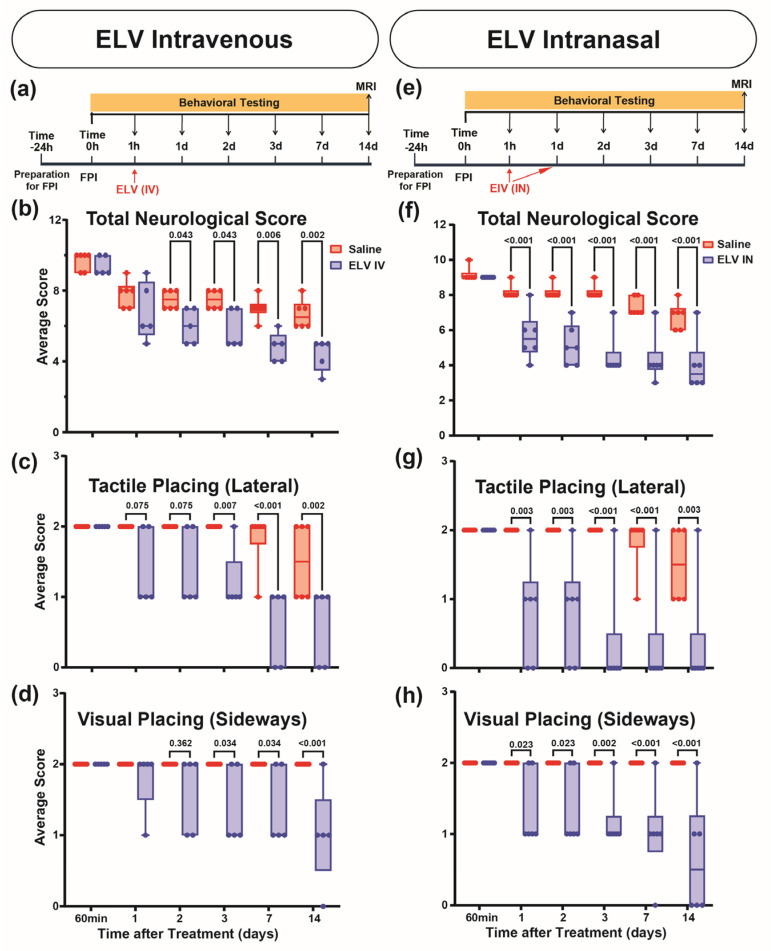
ELV-IV and ELV-IN remarkably improved the total neurological score and tactile lateral and visual sideways contralateral forelimb placing reactions 14 days after FPI. (**a**) ELV-IV: the experimental design. (**b**) The total neurological score was enhanced by ELV-IV (normal score = 0, max = 12). (**c**,**d**) Tactile lateral and visual sideways placing were improved by ELV-IV (normal = 0, max = 2). n = 5–6 per group. (**e**) ELV-IN experimental design. (**f**) ELV-IN also enhanced the total neurological score. (**g**) Tactile placing (lateral) and (**h**) visual placing (sideways) exhibited remarkable recoveries after ELV-IN. The values shown are mean ± SEM, n = 6–7 per group; multiple Mann-Whitney tests. In the graphs (**b**,**d**,**f**,**h**), red represents vehicle, and blue is ELV.

**Figure 3 biomedicines-12-02555-f003:**
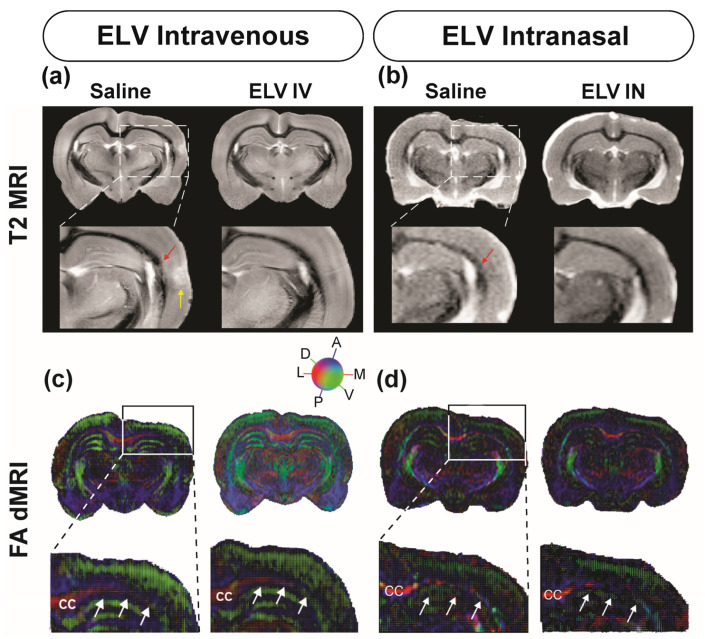
ELV-IV and ELV-IN attenuated brain damage and protected the integrity of the WM 14 days after FPI. (**a**) Representative T2WI images at the level of the dorsal hippocampus from saline and ELV-IV-treated rats on day 14. Saline rats exhibited more edema in the cortex (yellow arrow) and small hemorrhages within the white matter of the external capsule (red arrow) compared to the ELV-treated rats. (**b**) T2WI from FPI rats treated with saline or ELV-IN. There was a reduction in edema and white matter abnormalities (red arrow) in ELV-IN-treated rats. The dotted line indicates the expanded area showing white matter (corpus callosum; CC) and hippocampus (**c**). Directionally encoded fractional anisotropy maps of the cortex and CC (white arrows) show improvements in water directionality after treatment of ELV-IV. (**d**) Similarly, treatment with ELV-IN resulted in the conservation of the CC (white arrows). Inset: a color-coded directionality sphere for water diffusion.

**Figure 4 biomedicines-12-02555-f004:**
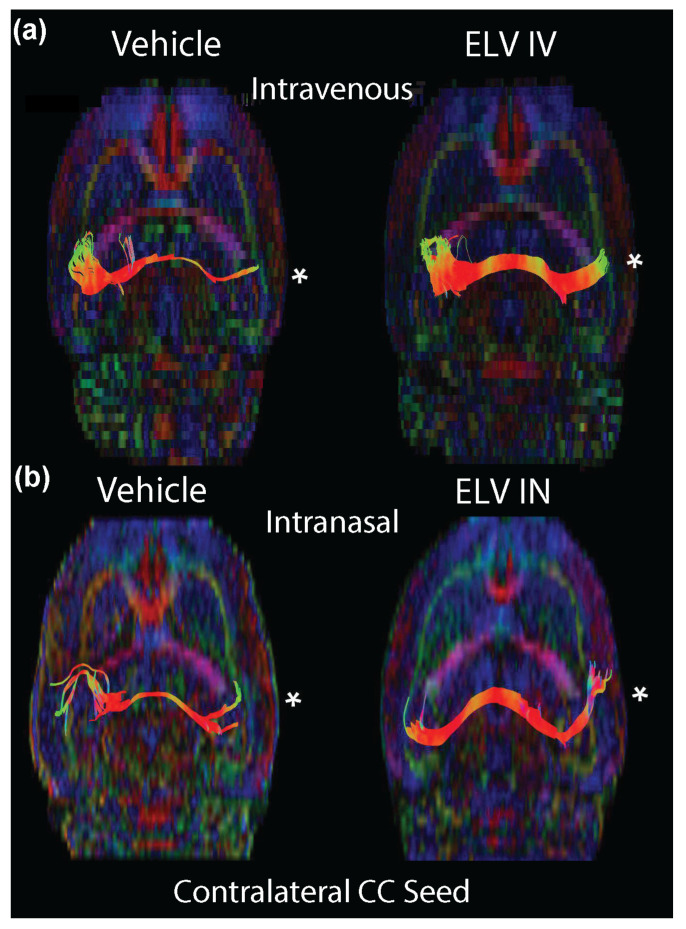
ELV-IV and ELV-IN preserved CC integrity. (**a**) CC tractography was examined using an ROI placed in the contralateral CC, and streamlines were visualized to the ipsilateral CC (injured hemisphere). ELV-IV treatment conferred protection of white matter tracts within CC. (**b**) ELV-IN treatment also increased the number of streamlines between the ipsi- and contralateral CC. * = ipsilateral FPI site of injury.

**Figure 5 biomedicines-12-02555-f005:**
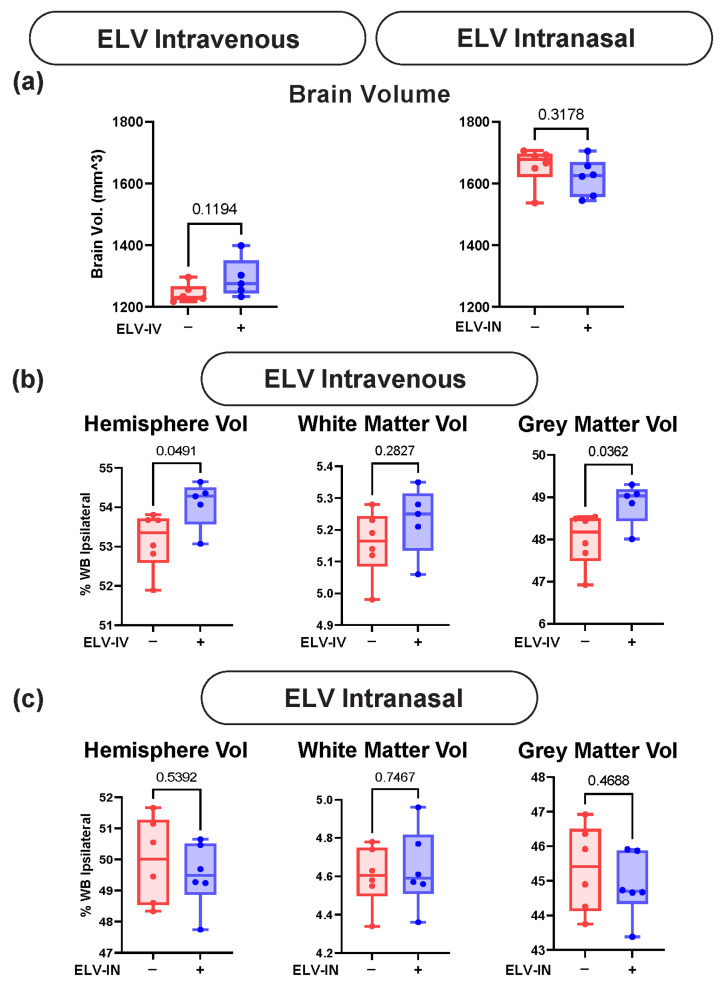
ELV treatment recovers regional brain volumes. (**a**) Whole-brain volume was not significantly altered after either ELV-IV or ELV-IN treatments. (**b**) ELV-IV exhibited significant increases in the ipsilateral hemisphere and gray matter volumes. (**c**) ELV-IN did not significantly alter white and gray matter or hemispheric volumes—Welch’s *t*-test.

**Figure 6 biomedicines-12-02555-f006:**
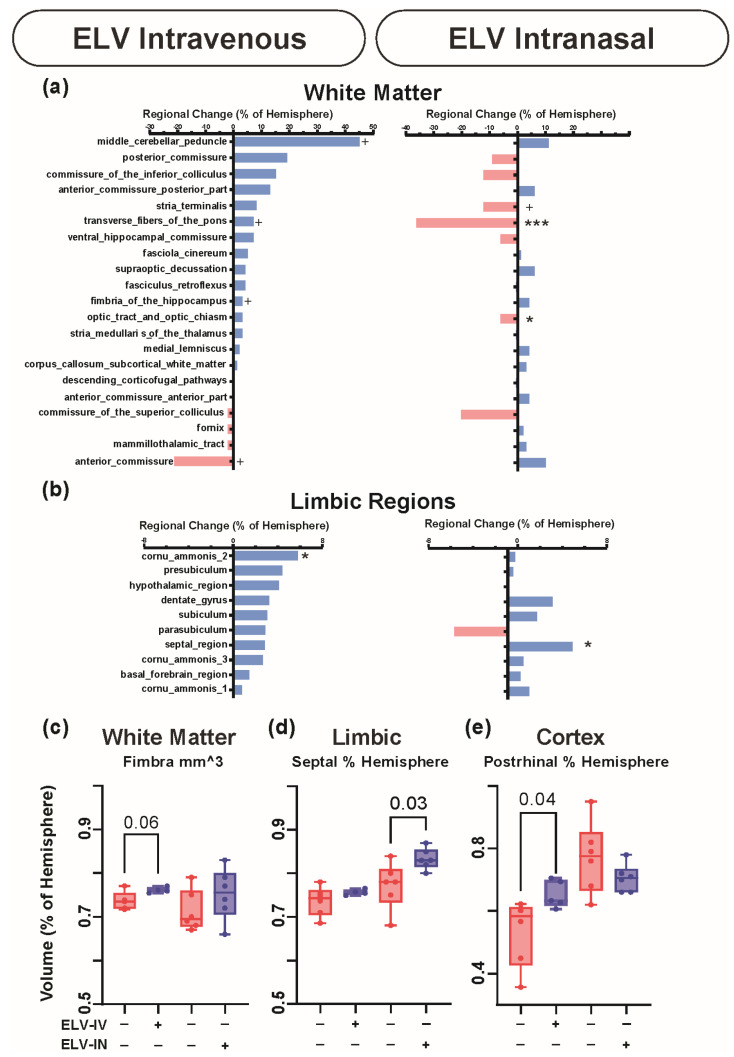
Regional volumetric sensitivity is different in Elavonoid treatments. (**a**) Regional white matter volume changes differ between ELV-IV and ELV-IN treatments. ELV-IV treatments broadly increased white matter volumes but ELV-IN treatment was less effective with fewer regions showing increased volumes. (**b**) Limbic regions had increased volumes with either treatment route, with ELV-IV appearing to be more effective. ((**a**,**b**) *t*-test: + = *p* < 0.1, * = *p* < 0.05, *** = *p* < 0.001). (**c**) The volume of the fimbra was increased (*p* = 0.06) after ELV-IV but not ELV-IN treatment. (**d**) Septum volume was not increased after ELV-IV treatment but significantly increased in ELV-IN (*p* = 0.03). (**e**) Cortical regions also exhibited increased volumes, with significant increases in postrhinal cortex after ELV-IV (*p* = 0.04) but not after ELV-IN treatment. *p* values *t*-test.

**Figure 7 biomedicines-12-02555-f007:**
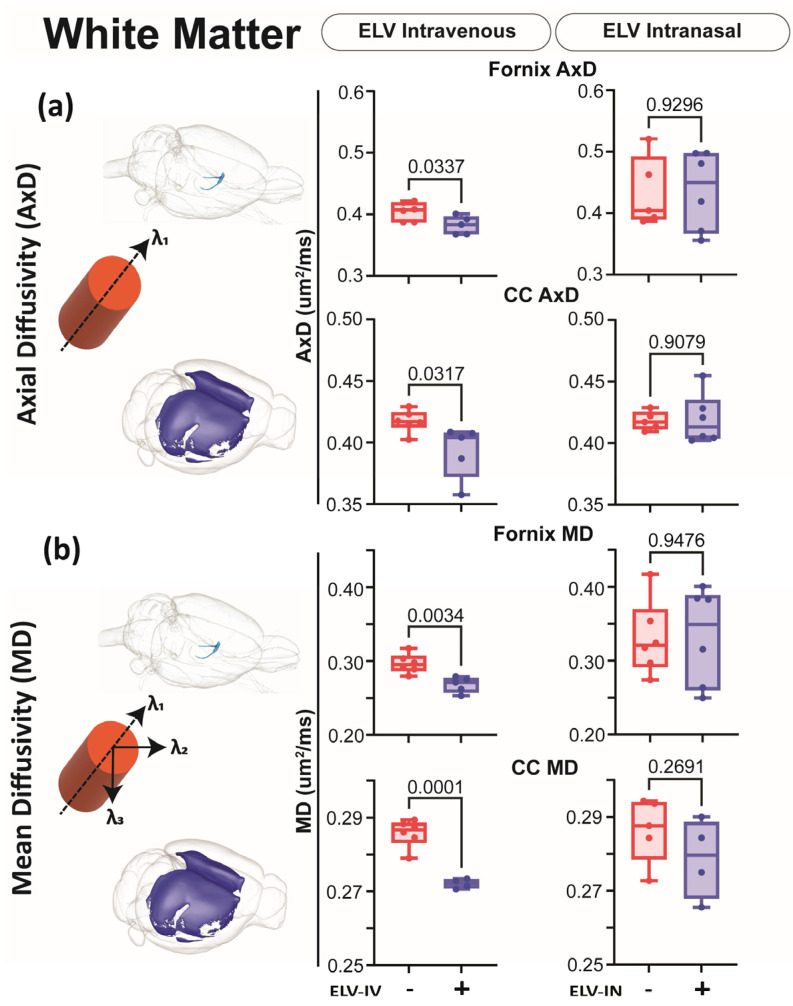
dMRI monitors improvements in white matter. (**a**) AxD diffusivity was reduced in the fornix and the CC after ELV-IV but not in ELV-IN-treated rats. (**b**) MD decreased significantly in ELV-IV rats compared to that in vehicles in both the fornix and CC but not in ELV-IN. In the graphs, red represents vehicle, and purple is ELV. Note the increased variance in ELV-IN rats—Welch’s *t*-test.

**Figure 8 biomedicines-12-02555-f008:**
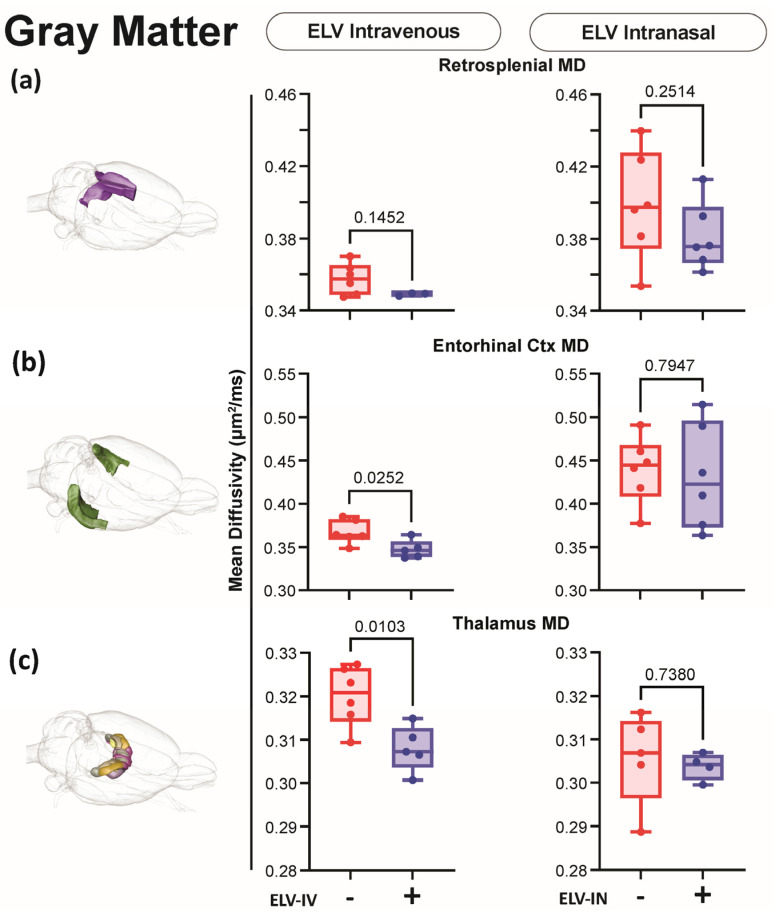
dMRI also reported improvements in gray matter in MD. (**a**) There was a trending decreased MD in the retrosplenial cortex in ELV-IV rats compared to saline ones, but no differences were observed for ELV-IN. (**b**) The MD was significantly reduced in the entorhinal cortex in ELV-IV but not ELV-IN rats. (**c**) The thalamus also reported significant reductions in MD in ELV-IV but not in ELV-IN rats—Welch’s *t*-test. In the graphs, red represents vehicle, and purple is ELV.

**Figure 9 biomedicines-12-02555-f009:**
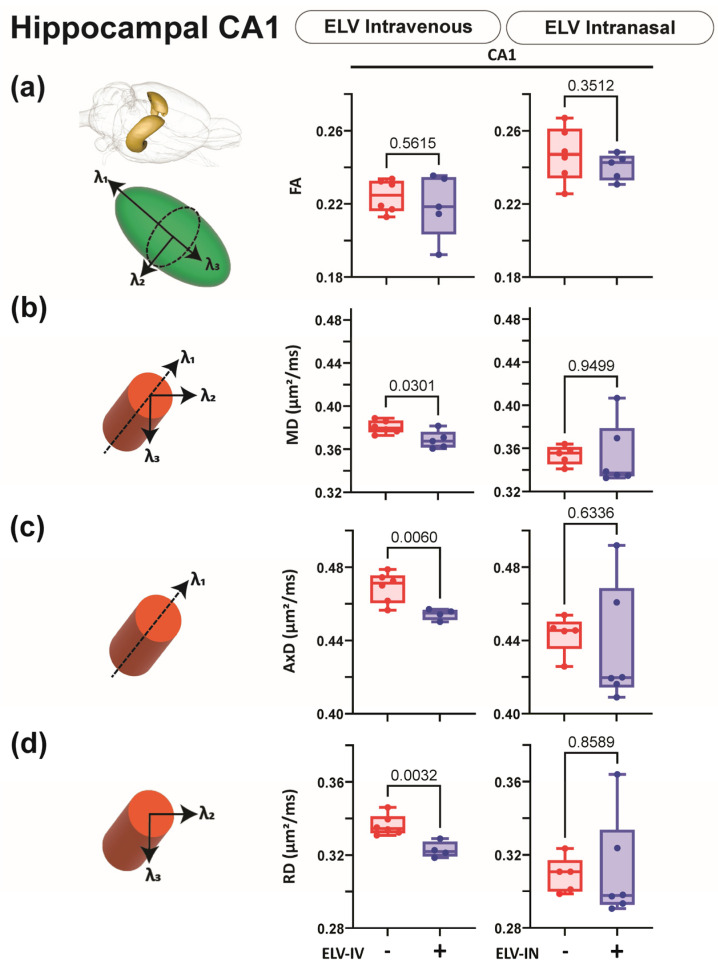
ELV-IV treatment results in improved CA1 dMRI. (**a**) FA was not significantly altered in either ELV-IV or -IN treatments. (**b**) The MD was reduced considerably in the CA1 region after ELV-IV treatment but not with ELV-IN treatment. (**c**) In the CA1 region, ELV-IV treatment resulted in significant reductions in AxD that were not observed in ELV-IN rats. (**d**) The RD in ELV-IV rats was significantly reduced, with no change in ELV-IN rats. In the graphs, red represents vehicle, and purple is ELV. Note the increased variance in the ELV-IN-treated rats for most dMRI metrics—Welch’s *t*-test.

**Figure 10 biomedicines-12-02555-f010:**
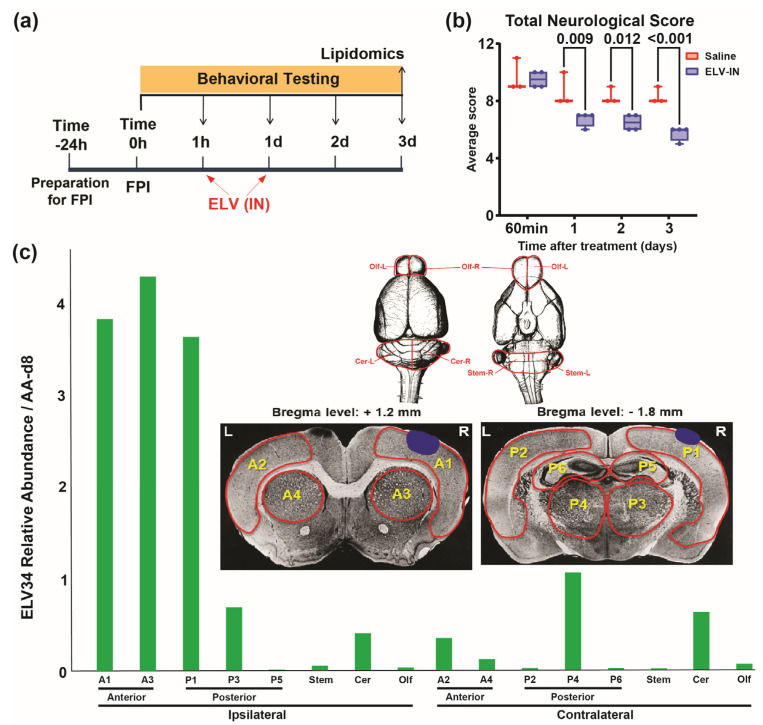
ELV-IN was detected in the brain 3 days after FPI. (**a**) Timeline for experimental procedures and tissue collection. Elov-Mix was delivered at 1 h and 24 h after FPI, and brain regions were sampled 3 days after FPI for lipid extraction and analysis. (**b**) Total neurological score (normal score = 0, max = 12). ELV-IN improved the total neurological score by 22, 22, and 31% compared to saline treatment on days 1, 2, and 3. Repeated-measures ANOVA followed by Bonferroni’s test. (**c**) Relative abundance of ELV-34 (ratio to internal standard AA-d8) in brain regions denoted in the inset. The blue dot is an area of impact. The graph represents the quantification of a representative animal. ELV-34 was found at the highest levels in the cortex and subcortex ipsilateral to the side of FPI. N = 3 per group. Region sampling diagram: A—anterior (bregma level +1.2 mm), P—posterior (bregma level −1.8 mm). Ipsilateral cortex (A1, A3, P1, P3), P5 (hippocampus), Olf, Cer, stem. Contralateral cortex (A2, A4, P2, P4), P6 (hippocampus), Olf, Cer, stem. Olf—olfactory tract, Cer—cerebellum, stem—brainstem.

## Data Availability

Data are contained within the article.

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
