# Peer review of "Elovanoids, a Novel Class of Lipid Mediators, Are Neuroprotective in a Traumatic Brain Injury Model in Rats"

_biomedicines, 2024, doi:10.3390/biomedicines12112555_

Round 1
Reviewer 1 Report
Comments and Suggestions for Authors
I appreciate the authors for presenting this novel research article emphasizing the neuroprotective effects of Elovanoids in an experimental model of traumatic brain injury. This is a well-designed and organized study. The results are robust and support the hypothesis. My comments are as follows:
Based on Figure 6, septum volume did not increase after ELV-IV treatment but significantly increased with ELV-IN. Figure 7 shows that dMRI monitors improvements in white matter, while Figure 8 reports dMRI also detects improvements in gray matter, particularly in MD diffusivity. Figure 9 indicates that ELV-IV treatment results in improved CA1 dMRI measurements. Do the authors suggest that ELV-IV and ELV-IN have different effects at different sites?
In Figure 10, ELV-IN improved the total neurological score by 22%, 22%, and 31% compared to saline treatment on days 1, 2, and 3, respectively. ELV-IV treatment improved total neurological scores on days 2, 3, 7, and 14 by 20%, 23%, 31%, and 34%, respectively, compared to saline treatment (Figure 2B). Additionally, ELV-IN improved the total neurological score by 31%, 37%, 45%, 41%, and 41% on days 1, 2, 3, 7, and 14, respectively (Figure 2F). However, these findings do not seem comparable with the MRI study. Could the authors please clarify?
Author Response
Comment 1: Based on Figure 6, septum volume did not increase after ELV-IV treatment but significantly increased with ELV-IN. Figure 7 shows that dMRI monitors improvements in white matter, while Figure 8 reports dMRI also detects improvements in gray matter, particularly in MD diffusivity. Figure 9 indicates that ELV-IV treatment results in improved CA1 dMRI measurements. Do the authors suggest that ELV-IV and ELV-IN have different effects at different sites?
Response 1: We thank the reviewer for noting these perceived inconsistencies. There are a variety of reasons that could account for the reported differences (see lines 541-547): "The differences between regional changes in the EVL-IV and ELV-IN groups may be attributed to 1) differences in the body weight of the rats between the two groups and 2) known differences between drug delivery to brain tissues, with IV considered more direct. Also, variations in IN delivery have been reported [45], and 3) MRI signatures were only extracted at 14d post-injury, and there may be differences in efficacy at different time points. Irrespective of these differences, we conclude that ELV via either route was efficacious in ameliorating behavioral and MRI perturbations." We included this paragraph in the discussion.
Comment 2: In Figure 10, ELV-IN improved the total neurological score by 22%, 22%, and 31% compared to saline treatment on days 1, 2, and 3, respectively. ELV-IV treatment improved total neurological scores on days 2, 3, 7, and 14 by 20%, 23%, 31%, and 34%, respectively, compared to saline treatment (Figure 2B). Additionally, ELV-IN improved the total neurological score by 31%, 37%, 45%, 41%, and 41% on days 1, 2, 3, 7, and 14, respectively (Figure 2F). However, these findings do not seem comparable with the MRI study. Could the authors please clarify?
Response 2: ELV-IV treatment (Fig. 2B) looks like it works better than ELV-IN (Fig. 2F) and ELV-IN (Fig. 10), but it was not statistically significant.
Reviewer 2 Report
Comments and Suggestions for Authors
Authors investigated Elovanoids, a novel class of homeostatic lipid mediators, for neuroprotective propoerties in an experimental model of traumatic brain injury. The manuscript is well written and supported by sound research.
I have few comments:
Authors used only male mice in the study.
Authors describe long-term effects of TBI but did test the animals over a period of few months.
Why was whole brain volume different in ELV-IV and ELV-IN. 1200 mm3 vs. 1600mm3
Author Response
Comment 1: Authors used only male mice in the study.
Response 1: It was an exploratory study only to assess the effect of ELV on the TBI model. We will use female rats in future studies.
Comment 2: Authors describe long-term effects of TBI but did test the animals over a period of few months.
Why was whole brain volume different in ELV-IV and ELV-IN: 1200 mm3 vs. 1600 mm3?
Response 2: Animals were allowed to survive only for 14 days. It's not considered long-term. We clarified this issue in the manuscript. The ELV-IV and -IN groups were two different body weights, which can explain different whole brain volumes. "Drugs Administration and Experimental Groups" method section has been updated to reflect different body weights in these studies.
Reviewer 3 Report
Comments and Suggestions for Authors
The manuscript ”Elovanoids, a novel class of homeostatic lipid mediators, are neuroprotective in an experimental model of traumatic brain injury,” authored by Nicolas G. Bazan et al., presented an interesting approach to using a novel class of phospholipid-derived mediators, elovanoids, in the management of TBI, in a rat model of fluid percussion injury.
- The Authors should consider revising the title, as it is too long and phrasy;
- English and punctuation need improvement throughout the manuscript; some of the sentences are hard to understand and atypical (for example, in the abstract "Male rats received a moderate fluid-percussion injury (FPI) model"; "sustain a TBI", line 34; "making it difficult to predict which individuals will develop long-lasting clinical sequelae", line 35; "have the most promising laboratory evidence for their neuro-restorative capacities in TBI", line 56; "ELV displays neuroprotective bioactivities in vitro and in vivo experimental ischemic stroke", line 65; "We demonstrated that they have improved behaviour, decreased lesion size 7 days after the experimental stroke model in rats, and protected retinal pigment epithelial cells and photoreceptors", line 68; "treatment with IN and IV ELV administration", line 79; [........]; "We have been shown that", line 385; and so on);
- The Introduction does not resume efficiently the essentials of TBI; it could be improved to be more clear and structured;
- Figure 1: a list of all the abbreviations should be added to the figure legend; the use of abbreviations should be consistent throughout the manuscript;
- line 80 - FPI was not defined;
- line 89 - no. and date of the Ethics Approval should be specified;
- line 91 - how many days exactly?
- line 142 - the description of the behavioural tests should be improved and more detailed;
- line 160 - was not clear about the moment MRIs were administered to the animals;
- Results: the text should be more specific regarding the significance of the data, p-values are often missing, as well as the statistical tests that were applied;
- The Authors should explain why They are discussing mild TBI while using a moderate TBI model; also, in the Discussion section the implication of elovanoids in TBI mechanisms, by relation to TBI and PCS symptoms needs more addressing;
- Some of the readers could perceive the language as a bit too enthusiastic (eg. "spectacular");
- The Discussion section would benefit from revising the content and the form of expressing the ideas; also, the inconsistent use of abbreviations makes the text hard to read;
- The Conclusion section should be revised to present clearer and more specific conclusions about the experiments and the results.
Good luck!
Author Response
Comment 1: The Authors should consider revising the title, as it is too long and phrasy;
Response 1: The title was revised, per reviewer recommendation: “Elovanoids, a novel class of lipid mediators, are neuroprotective in traumatic brain injury model in rats.”
Comment 2: English and punctuation need improvement throughout the manuscript; some of the sentences are hard to understand and atypical (for example, in the abstract "Male rats received a moderate fluid-percussion injury (FPI) model"; "sustain a TBI", line 34; "making it difficult to predict which individuals will develop long-lasting clinical sequelae", line 35; "have the most promising laboratory evidence for their neuro-restorative capacities in TBI", line 56; "ELV displays neuroprotective bioactivities in vitro and in vivo experimental ischemic stroke", line 65; "We demonstrated that they have improved behaviour, decreased lesion size 7 days after the experimental stroke model in rats, and protected retinal pigment epithelial cells and photoreceptors", line 68; "treatment with IN and IV ELV administration", line 79; [........]; "We have been shown that", line 385; and so on);
Response 2: We have improved the English and punctuation throughout the manuscript, as suggested.
Comment 3: The Introduction does not resume efficiently the essentials of TBI; it could be improved to be more clear and structured;
Response 3: As the reviewer recommended, the following paragraph is included in the introduction: “TBI can be divided into three categories: closed head, penetrating, and explosive blast TBI [2]. Closed-head TBI is typically caused by blunt impact incurred mainly from motor vehicle accidents, falls, and sports activities. The incidence rate of this form of TBI is the highest amongst the civilian population. The strong blunt and compression contact force disrupts the brain's normal functioning directly underneath the site of impact, thus causing immediate damage to brain vasculature and neuronal cells. Mild-to-moderate TBI accounts for about 90% of all TBIs, while approximately 80% of TBI cases in the United States are classified as mild (mTBI) [2,3].”
Comment 4: Figure 1: a list of all the abbreviations should be added to the figure legend; the use of abbreviations should be consistent throughout the manuscript;
Response 4: A full list of abbreviations has now been added to the Figure 1 legend.
Comment 5: line 80 - FPI was not defined;
Response 5: The definition has now been added.
Comment 6: line 89 - no. and date of the Ethics Approval should be specified;
Response 6: The date is now included: 07/22/2024.
Comment 7: line 91 - how many days exactly?
Response 7: This has been corrected to “three days” in line 115.
Comment 8: line 142 - the description of the behavioural tests should be improved and more detailed;
Response 8: Detailed behavioral tests are now included in 2.4. Behavioral Tests (lines 177-204).
Comment 9: line 160 - was not clear about the moment MRIs were administered to the animals;
Response 9: 2.6. Magnetic Resonance Imaging (MRI) Acquisition and Analysis was updated to include: “MRI was conducted on day 14 after TBI.”
Comment 10: Results: the text should be more specific regarding the significance of the data, p-values are often missing, as well as the statistical tests that were applied;
Response 10: We included p-values and statistical tests in the legends to better understand the results.
Comment 11: The Authors should explain why They are discussing mild TBI while using a moderate TBI model; also, in the Discussion section the implication of elovanoids in TBI mechanisms, by relation to TBI and PCS symptoms needs more addressing;
Response 11: We decided to include mild TBI because it very often can lead to moderate TBI if untreated. We addressed PCS and mechanisms of ELV neuroprotection in the discussion. We included the requested information in the Discussion section (lines 437-460).
Comment 12: Some of the readers could perceive the language as a bit too enthusiastic (eg. "spectacular");
The Discussion section would benefit from revising the content and the form of expressing the ideas; also, the inconsistent use of abbreviations makes the text hard to read;
Response 12: Thank you for pointing this out. "Spectacular" has now been replaced by "remarkable," and we have cleaned up the abbreviations to make them more consistent.
Comment 13: The Conclusion section should be revised to present clearer and more specific conclusions about the experiments and the results.
Response 13: We demonstrated here that both deliveries of ELV-IV and ELV-IN attenuated brain damage by improving behavior, protecting the integrity of the WM, preserving CC integrity, recovering brain injury volumes, and improving white and gray matter diffusivity. We have also shown that ELV can be selectively delivered intranasally, as detected in the brain by lipidomic analysis.
Round 2
Reviewer 3 Report
Comments and Suggestions for Authors
The manuscript "Elovanoids, a novel class of homeostatic lipid mediators, are neuroprotective in an experimental model of traumatic brain injury" by Bazan et al. improved its quality after revision.
Author Response
Comment 1: The manuscript "Elovanoids, a novel class of homeostatic lipid mediators, are neuroprotective in an experimental model of traumatic brain injury" by Bazan et al. improved its quality after revision.
Response 1: Thank you for helping us improve the manuscript.